# ON UNIFYING DEEP GENERATIVE MODELS

**Zhiting Hu**[1,2]    **Zichao Yang**[1]    **Ruslan Salakhutdinov**[1]    **Eric P. Xing**[1,2]
Carnegie Mellon University[1],  Petuum Inc.[2]

## ABSTRACT

Deep generative models have achieved impressive success in recent years. Generative Adversarial Networks (GANs) and Variational Autoencoders (VAEs), as powerful frameworks for deep generative model learning, have largely been considered as two distinct paradigms and received extensive independent studies respectively. This paper aims to establish formal connections between GANs and VAEs through a new formulation of them. We interpret sample generation in GANs as performing posterior inference, and show that GANs and VAEs involve minimizing KL divergences of respective posterior and inference distributions with opposite directions, extending the two learning phases of classic wake-sleep algorithm, respectively. The unified view provides a powerful tool to analyze a diverse set of existing model variants, and enables to transfer techniques across research lines in a principled way. For example, we apply the importance weighting method in VAE literatures for improved GAN learning, and enhance VAEs with an adversarial mechanism that leverages generated samples. Experiments show generality and effectiveness of the transfered techniques.

## 1 INTRODUCTION

Deep generative models define distributions over a set of variables organized in multiple layers. Early forms of such models dated back to works on hierarchical Bayesian models (Neal, 1992) and neural network models such as Helmholtz machines (Dayan et al., 1995), originally studied in the context of unsupervised learning, latent space modeling, etc. Such models are usually trained via an EM style framework, using either a variational inference (Jordan et al., 1999) or a data augmentation (Tanner & Wong, 1987) algorithm. Of particular relevance to this paper is the classic wake-sleep algorithm dates by Hinton et al. (1995) for training Helmholtz machines, as it explored an idea of minimizing a pair of KL divergences in opposite directions of the posterior and its approximation.

In recent years there has been a resurgence of interests in deep generative modeling. The emerging approaches, including Variational Autoencoders (VAEs) (Kingma & Welling, 2013), Generative Adversarial Networks (GANs) (Goodfellow et al., 2014), Generative Moment Matching Networks (GMMNs) (Li et al., 2015; Dziugaite et al., 2015), auto-regressive neural networks (Larochelle & Murray, 2011; Oord et al., 2016), and so forth, have led to impressive results in a myriad of applications, such as image and text generation (Radford et al., 2015; Hu et al., 2017; van den Oord et al., 2016), disentangled representation learning (Chen et al., 2016; Kulkarni et al., 2015), and semi-supervised learning (Salimans et al., 2016; Kingma et al., 2014).

The deep generative model literature has largely viewed these approaches as distinct model training paradigms. For instance, GANs aim to achieve an equilibrium between a generator and a discriminator; while VAEs are devoted to maximizing a variational lower bound of the data log-likelihood. A rich array of theoretical analyses and model extensions have been developed independently for GANs (Arjovsky & Bottou, 2017; Arora et al., 2017; Salimans et al., 2016; Nowozin et al., 2016) and VAEs (Burda et al., 2015; Chen et al., 2017; Hu et al., 2017), respectively. A few works attempt to combine the two objectives in a single model for improved inference and sample generation (Mescheder et al., 2017; Larsen et al., 2015; Makhzani et al., 2015; Sønderby et al., 2017). Despite the significant progress specific to each method, it remains unclear how these apparently divergent approaches connect to each other in a principled way.

In this paper, we present a new formulation of GANs and VAEs that connects them under a unified view, and links them back to the classic wake-sleep algorithm. We show that GANs and VAEs

involve minimizing opposite KL divergences of respective posterior and inference distributions, and extending the sleep and wake phases, respectively, for generative model learning. More specifically, we develop a reformulation of GANs that interprets *generation* of samples as performing posterior *inference*, leading to an objective that resembles variational inference as in VAEs. As a counterpart, VAEs in our interpretation contain a *degenerated* adversarial mechanism that blocks out generated samples and only allows real examples for model training.

The proposed interpretation provides a useful tool to analyze the broad class of recent GAN- and VAE-based algorithms, enabling perhaps a more principled and unified view of the landscape of generative modeling. For instance, one can easily extend our formulation to subsume InfoGAN (Chen et al., 2016) that additionally infers hidden representations of examples, VAE/GAN joint models (Larsen et al., 2015; Che et al., 2017a) that offer improved generation and reduced mode missing, and adversarial domain adaptation (ADA) (Ganin et al., 2016; Purushotham et al., 2017) that is traditionally framed in the discriminative setting.

The close parallelisms between GANs and VAEs further ease transferring techniques that were originally developed for improving each individual class of models, to in turn benefit the other class. We provide two examples in such spirit: 1) Drawn inspiration from importance weighted VAE (IWAE) (Burda et al., 2015), we straightforwardly derive importance weighted GAN (IWGAN) that maximizes a tighter lower bound on the marginal likelihood compared to the vanilla GAN. 2) Motivated by the GAN adversarial game we activate the originally degenerated discriminator in VAEs, resulting in a full-fledged model that adaptively leverages both real and fake examples for learning. Empirical results show that the techniques imported from the other class are generally applicable to the base model and its variants, yielding consistently better performance.

## 2 RELATED WORK

There has been a surge of research interest in deep generative models in recent years, with remarkable progress made in understanding several class of algorithms. The wake-sleep algorithm (Hinton et al., 1995) is one of the earliest general approaches for learning deep generative models. The algorithm incorporates a separate inference model for posterior approximation, and aims at maximizing a variational lower bound of the data log-likelihood, or equivalently, minimizing the KL divergence of the approximate posterior and true posterior. However, besides the wake phase that minimizes the KL divergence w.r.t the generative model, the sleep phase is introduced for tractability that minimizes instead the *reversed* KL divergence w.r.t the inference model. Recent approaches such as NVIL (Mnih & Gregor, 2014) and VAEs (Kingma & Welling, 2013) are developed to maximize the variational lower bound w.r.t both the generative and inference models jointly. To reduce the variance of stochastic gradient estimates, VAEs leverage reparametrized gradients. Many works have been done along the line of improving VAEs. Burda et al. (2015) develop importance weighted VAEs to obtain a tighter lower bound. As VAEs do not involve a sleep phase-like procedure, the model cannot leverage samples from the generative model for model training. Hu et al. (2017) combine VAEs with an extended sleep procedure that exploits generated samples for learning.

Another emerging family of deep generative models is the Generative Adversarial Networks (GANs) (Goodfellow et al., 2014), in which a discriminator is trained to distinguish between real and generated samples and the generator to confuse the discriminator. The adversarial approach can be alternatively motivated in the perspectives of approximate Bayesian computation (Gutmann et al., 2014) and density ratio estimation (Mohamed & Lakshminarayanan, 2016). The original objective of the generator is to minimize the log probability of the discriminator correctly recognizing a generated sample as fake. This is equivalent to *minimizing a lower bound* on the Jensen-Shannon divergence (JSD) of the generator and data distributions (Goodfellow et al., 2014; Nowozin et al., 2016; Huszar, 2016; Li, 2016). Besides, the objective suffers from vanishing gradient with strong discriminator. Thus in practice people have used another objective which maximizes the log probability of the discriminator recognizing a generated sample as real (Goodfellow et al., 2014; Arjovsky & Bottou, 2017). The second objective has the same optimal solution as with the original one. We base our analysis of GANs on the second objective as it is widely used in practice yet few theoretic analysis has been done on it. Numerous extensions of GANs have been developed, including combination with VAEs for improved generation (Larsen et al., 2015; Makhzani et al., 2015; Che et al., 2017a), and generalization of the objectives to minimize other f-divergence criteria beyond JSD (Nowozin

et al., 2016; Sønderby et al., 2017). The adversarial principle has gone beyond the generation setting and been applied to other contexts such as domain adaptation (Ganin et al., 2016; Purushotham et al., 2017), and Bayesian inference (Mescheder et al., 2017; Tran et al., 2017; Huszár, 2017; Rosca et al., 2017) which uses implicit variational distributions in VAEs and leverage the adversarial approach for optimization. This paper starts from the basic models of GANs and VAEs, and develops a general formulation that reveals underlying connections of different classes of approaches including many of the above variants, yielding a unified view of the broad set of deep generative modeling.

## 3 Bridging the Gap

The structures of GANs and VAEs are at the first glance quite different from each other. VAEs are based on the variational inference approach, and include an explicit inference model that reverses the generative process defined by the generative model. On the contrary, in traditional view GANs lack an inference model, but instead have a discriminator that judges generated samples. In this paper, a key idea to bridge the gap is to interpret the generation of samples in GANs as *performing inference*, and the discrimination as a generative process that produces real/fake labels. The resulting new formulation reveals the connections of GANs to traditional variational inference. The reversed generation-inference interpretations between GANs and VAEs also expose their correspondence to the two learning phases in the classic wake-sleep algorithm.

For ease of presentation and to establish a systematic notation for the paper, we start with a new interpretation of *Adversarial Domain Adaptation* (ADA) (Ganin et al., 2016), the application of adversarial approach in the domain adaptation context. We then show GANs are a special case of ADA, followed with a series of analysis linking GANs, VAEs, and their variants in our formulation.

### 3.1 Adversarial Domain Adaptation (ADA)

ADA aims to transfer prediction knowledge learned from a source domain to a target domain, by learning domain-invariant features (Ganin et al., 2016). That is, it learns a feature extractor whose output cannot be distinguished by a discriminator between the source and target domains.

We first review the conventional formulation of ADA. Figure 1(a) illustrates the computation flow. Let $z$ be a data example either in the source or target domain, and $y \in \{0, 1\}$ the domain indicator with $y = 0$ indicating the target domain and $y = 1$ the source domain. The data distributions conditioning on the domain are then denoted as $p(z|y)$. The feature extractor $G_\theta$ parameterized with $\theta$ maps $z$ to feature $x = G_\theta(z)$. To enforce domain invariance of feature $x$, a discriminator $D_\phi$ is learned. Specifically, $D_\phi(x)$ outputs the probability that $x$ comes from the source domain, and the discriminator is trained to maximize the binary classification accuracy of recognizing the domains:

$$\max_\phi \mathcal{L}_\phi = \mathbb{E}_{x=G_\theta(z),z\sim p(z|y=1)}\left[\log D_\phi(x)\right] + \mathbb{E}_{x=G_\theta(z),z\sim p(z|y=0)}\left[\log(1 - D_\phi(x))\right]. \quad (1)$$

The feature extractor $G_\theta$ is then trained to fool the discriminator:

$$\max_\theta \mathcal{L}_\theta = \mathbb{E}_{x=G_\theta(z),z\sim p(z|y=1)}\left[\log(1 - D_\phi(x))\right] + \mathbb{E}_{x=G_\theta(z),z\sim p(z|y=0)}\left[\log D_\phi(x)\right]. \quad (2)$$

Please see the supplementary materials for more details of ADA.

With the background of conventional formulation, we now frame our new interpretation of ADA. The data distribution $p(z|y)$ and deterministic transformation $G_\theta$ together form an *implicit* distribution over $x$, denoted as $p_\theta(x|y)$, which is intractable to evaluate likelihood but easy to sample from. Let $p(y)$ be the distribution of the domain indicator $y$, e.g., a uniform distribution as in Eqs.(1)-(2). The discriminator defines a conditional distribution $q_\phi(y|x) = D_\phi(x)$. Let $q_\phi^r(y|x) = q_\phi(1 - y|x)$ be the reversed distribution over domains. The objectives of ADA are therefore rewritten as (omitting the constant scale factor 2):

$$\begin{aligned}\max_\phi \mathcal{L}_\phi &= \mathbb{E}_{p_\theta(x|y)p(y)}\left[\log q_\phi(y|x)\right] \\ \max_\theta \mathcal{L}_\theta &= \mathbb{E}_{p_\theta(x|y)p(y)}\left[\log q_\phi^r(y|x)\right].\end{aligned} \quad (3)$$

Note that $z$ is encapsulated in the implicit distribution $p_\theta(x|y)$. The only difference of the objectives of $\theta$ from $\phi$ is the replacement of $q(y|x)$ with $q^r(y|x)$. This is where the adversarial mechanism comes about. We defer deeper interpretation of the new objectives in the next subsection.

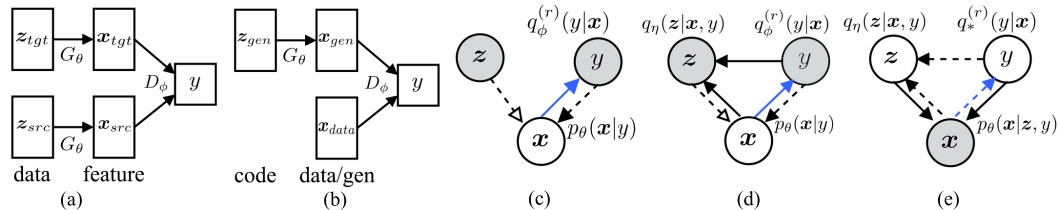

Figure 1: **(a)** Conventional view of ADA. To make direct correspondence to GANs, we use $z$ to denote the data and $x$ the feature. Subscripts *src* and *tgt* denote source and target domains, respectively. **(b)** Conventional view of GANs. **(c)** Schematic graphical model of both ADA and GANs (Eq.3). Arrows with solid lines denote generative process; arrows with dashed lines denote inference; hollow arrows denote deterministic transformation leading to implicit distributions; and blue arrows denote adversarial mechanism that involves respective conditional distribution $q$ and its reverse $q^r$, e.g., $q(y|x)$ and $q^r(y|x)$ (denoted as $q^{(r)}(y|x)$ for short). Note that in GANs we have interpreted $x$ as latent variable and $(z, y)$ as visible. **(d)** InfoGAN (Eq.9), which, compared to GANs, adds conditional generation of code $z$ with distribution $q_\eta(z|x, y)$. **(e)** VAEs (Eq.12), which is obtained by swapping the generation and inference processes of InfoGAN, i.e., in terms of the schematic graphical model, swapping solid-line arrows (generative process) and dashed-line arrows (inference) of (d).

## 3.2 GENERATIVE ADVERSARIAL NETWORKS (GANS)

GANs (Goodfellow et al., 2014) can be seen as a special case of ADA. Taking image generation for example, intuitively, we want to transfer the properties of real image (source domain) to generated image (target domain), making them indistinguishable to the discriminator. Figure 1(b) shows the conventional view of GANs.

Formally, $x$ now denotes a real example or a generated sample, $z$ is the respective latent code. For the generated sample domain ($y = 0$), the implicit distribution $p_\theta(x|y = 0)$ is defined by the prior of $z$ and the generator $G_\theta(z)$, which is also denoted as $p_{g_\theta}(x)$ in the literature. For the real example domain ($y = 1$), the code space and generator are *degenerated*, and we are directly presented with a fixed distribution $p(x|y = 1)$, which is just the real data distribution $p_{data}(x)$. Note that $p_{data}(x)$ is also an implicit distribution and allows efficient empirical sampling. In summary, the conditional distribution over $x$ is constructed as

$$p_\theta(x|y) = \begin{cases} p_{g_\theta}(x) & y = 0 \\ p_{data}(x) & y = 1. \end{cases} \quad (4)$$

Here, free parameters $\theta$ are only associated with $p_{g_\theta}(x)$ of the generated sample domain, while $p_{data}(x)$ is constant. As in ADA, discriminator $D_\phi$ is simultaneously trained to infer the probability that $x$ comes from the real data domain. That is, $q_\phi(y = 1|x) = D_\phi(x)$.

With the established correspondence between GANs and ADA, we can see that the objectives of GANs are precisely expressed as Eq.(3). To make this clearer, we recover the classical form by unfolding over $y$ and plugging in conventional notations. For instance, the objective of the generative parameters $\theta$ in Eq.(3) is translated into

$$\max_\theta \mathcal{L}_\theta = \mathbb{E}_{p_\theta(x|y=0)p(y=0)} \left[ \log q_\phi^r(y = 0|x) \right] + \mathbb{E}_{p_\theta(x|y=1)p(y=1)} \left[ \log q_\phi^r(y = 1|x) \right]$$
$$= \frac{1}{2}\mathbb{E}_{x=G_\theta(z), z\sim p(z|y=0)} \left[ \log D_\phi(x) \right] + const, \quad (5)$$

where $p(y)$ is uniform and results in the constant scale factor $1/2$. As noted in sec.2, we focus on the unsaturated objective for the generator (Goodfellow et al., 2014), as it is commonly used in practice yet still lacks systematic analysis.

**New Interpretation** Let us take a closer look into the form of Eq.(3). It closely resembles the data reconstruction term of a variational lower bound by treating $y$ as visible variable while $x$ as latent (as in ADA). That is, we are essentially reconstructing the real/fake indicator $y$ (or its reverse $1 - y$) with the "generative distribution" $q_\phi(y|x)$ and conditioning on $x$ from the "inference distribution" $p_\theta(x|y)$. Figure 1(c) shows a schematic graphical model that illustrates such generative and inference processes. (Sec.D in the supplementary materials gives an example of translating a given schematic graphical model into mathematical formula.) We go a step further to reformulate the objectives and reveal more insights to the problem. In particular, for each optimization step of $p_\theta(x|y)$ at point $(\theta_0, \phi_0)$ in the parameter space, we have:

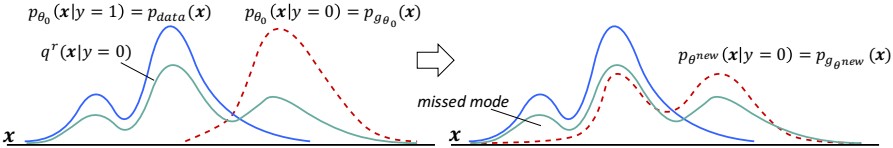

Figure 2: One optimization step of the parameter $\boldsymbol{\theta}$ through Eq.(6) at point $\boldsymbol{\theta}_0$. The posterior $q^r(\boldsymbol{x}|y)$ is a mixture of $p_{\theta_0}(\boldsymbol{x}|y=0)$ (blue) and $p_{\theta_0}(\boldsymbol{x}|y=1)$ (red in the left panel) with the mixing weights induced from $q_{\phi_0}^r(y|\boldsymbol{x})$. Minimizing the KLD drives $p_\theta(\boldsymbol{x}|y=0)$ towards the respective mixture $q^r(\boldsymbol{x}|y=0)$ (green), resulting in a new state where $p_{\theta^{new}}(\boldsymbol{x}|y=0) = p_{g_{\theta^{new}}}(\boldsymbol{x})$ (red in the right panel) gets closer to $p_{\theta_0}(\boldsymbol{x}|y=1) = p_{data}(\boldsymbol{x})$. Due to the asymmetry of KLD, $p_{g_{\theta^{new}}}(\boldsymbol{x})$ missed the smaller mode of the mixture $q^r(\boldsymbol{x}|y=0)$ which is a mode of $p_{data}(\boldsymbol{x})$.

**Lemma 1.** *Let $p(y)$ be the uniform distribution. Let $p_{\theta_0}(\boldsymbol{x}) = \mathbb{E}_{p(y)}[p_{\theta_0}(\boldsymbol{x}|y)]$, and $q^r(\boldsymbol{x}|y) \propto q_{\phi_0}^r(y|\boldsymbol{x})p_{\theta_0}(\boldsymbol{x})$. Therefore, the updates of $\boldsymbol{\theta}$ at $\boldsymbol{\theta}_0$ have*

$$\nabla_\theta \Big[ - \mathbb{E}_{p_\theta(\boldsymbol{x}|y)p(y)} \big[ \log q_{\phi_0}^r(y|\boldsymbol{x}) \big] \Big] \Big|_{\boldsymbol{\theta}=\boldsymbol{\theta}_0} =$$
$$\nabla_\theta \Big[ \mathbb{E}_{p(y)} \big[ KL \left( p_\theta(\boldsymbol{x}|y) \big\| q^r(\boldsymbol{x}|y) \right) \big] - JSD \left( p_\theta(\boldsymbol{x}|y=0) \big\| p_\theta(\boldsymbol{x}|y=1) \right) \Big] \Big|_{\boldsymbol{\theta}=\boldsymbol{\theta}_0}, \tag{6}$$

*where $KL(\cdot\|\cdot)$ and $JSD(\cdot\|\cdot)$ are the KL and Jensen-Shannon Divergences, respectively.*

Proofs are in the supplements (sec.B). Eq.(6) offers several insights into the GAN generator learning:

- **Resemblance to variational inference.** As above, we see $\boldsymbol{x}$ as latent and $p_\theta(\boldsymbol{x}|y)$ as the inference distribution. The $p_{\theta_0}(\boldsymbol{x})$ is fixed to the starting state of the current update step, and can naturally be seen as the prior over $\boldsymbol{x}$. By definition $q^r(\boldsymbol{x}|y)$ that combines the prior $p_{\theta_0}(\boldsymbol{x})$ and the generative distribution $q_{\phi_0}^r(y|\boldsymbol{x})$ thus serves as the posterior. Therefore, optimizing the generator $G_\theta$ is equivalent to minimizing the KL divergence between the inference distribution and the posterior (a standard from of variational inference), minus a JSD between the distributions $p_{g_\theta}(\boldsymbol{x})$ and $p_{data}(\boldsymbol{x})$. The interpretation further reveals the connections to VAEs, as discussed later.

- **Training dynamics.** By definition, $p_{\theta_0}(\boldsymbol{x}) = (p_{g_{\theta_0}}(\boldsymbol{x}) + p_{data}(\boldsymbol{x}))/2$ is a mixture of $p_{g_{\theta_0}}(\boldsymbol{x})$ and $p_{data}(\boldsymbol{x})$ with uniform mixing weights, so the posterior $q^r(\boldsymbol{x}|y) \propto q_{\phi_0}^r(y|\boldsymbol{x})p_{\theta_0}(\boldsymbol{x})$ is also a mixture of $p_{g_{\theta_0}}(\boldsymbol{x})$ and $p_{data}(\boldsymbol{x})$ with mixing weights induced from the discriminator $q_{\phi_0}^r(y|\boldsymbol{x})$. For the KL divergence to minimize, the component with $y=1$ is $KL\left(p_\theta(\boldsymbol{x}|y=1)\|q^r(\boldsymbol{x}|y=1)\right) = KL\left(p_{data}(\boldsymbol{x})\|q^r(\boldsymbol{x}|y=1)\right)$ which is a constant. The active component for optimization is with $y=0$, i.e., $KL\left(p_\theta(\boldsymbol{x}|y=0)\|q^r(\boldsymbol{x}|y=0)\right) = KL\left(p_{g_\theta}(\boldsymbol{x})\|q^r(\boldsymbol{x}|y=0)\right)$. Thus, minimizing the KL divergence in effect drives $p_{g_\theta}(\boldsymbol{x})$ to a mixture of $p_{g_{\theta_0}}(\boldsymbol{x})$ and $p_{data}(\boldsymbol{x})$. Since $p_{data}(\boldsymbol{x})$ is fixed, $p_{g_\theta}(\boldsymbol{x})$ gets closer to $p_{data}(\boldsymbol{x})$. Figure 2 illustrates the training dynamics schematically.

- **The JSD term.** The negative JSD term is due to the introduction of the prior $p_{\theta_0}(\boldsymbol{x})$. This term pushes $p_{g_\theta}(\boldsymbol{x})$ away from $p_{data}(\boldsymbol{x})$, which acts oppositely from the KLD term. However, we show that the JSD term is upper bounded by the KLD term (sec.C). Thus, if the KLD term is sufficiently minimized, the magnitude of the JSD also decreases. Note that we do not mean the JSD is insignificant or negligible. Instead conclusions drawn from Eq.(6) should take the JSD term into account.

- **Explanation of missing mode issue.** JSD is a symmetric divergence measure while KLD is non-symmetric. The missing mode behavior widely observed in GANs (Metz et al., 2017; Che et al., 2017a) is thus explained by the asymmetry of the KLD which tends to concentrate $p_\theta(\boldsymbol{x}|y)$ to large modes of $q^r(\boldsymbol{x}|y)$ and ignore smaller ones. See Figure 2 for the illustration. Concentration to few large modes also facilitates GANs to generate sharp and realistic samples.

- **Optimality assumption of the discriminator.** Previous theoretical works have typically assumed (near) optimal discriminator (Goodfellow et al., 2014; Arjovsky & Bottou, 2017):

$$q_{\phi_0}(y|\boldsymbol{x}) \approx \frac{p_{\theta_0}(\boldsymbol{x}|y=1)}{p_{\theta_0}(\boldsymbol{x}|y=0) + p_{\theta_0}(\boldsymbol{x}|y=1)} = \frac{p_{data}(\boldsymbol{x})}{p_{g_{\theta_0}}(\boldsymbol{x}) + p_{data}(\boldsymbol{x})}, \tag{7}$$

which can be unwarranted in practice due to limited expressiveness of the discriminator (Arora et al., 2017). In contrast, our result does not rely on the optimality assumptions. Indeed, our result is a generalization of the previous theorem in (Arjovsky & Bottou, 2017), which is recovered by

plugging Eq.(7) into Eq.(6):

$$\nabla_\theta \Big[ - \mathbb{E}_{p_\theta(\boldsymbol{x}|y)p(y)} \big[ \log q_{\phi_0}^r(y|\boldsymbol{x}) \big] \Big] \Big|_{\boldsymbol{\theta}=\boldsymbol{\theta}_0} = \nabla_\theta \Big[ \frac{1}{2} \mathrm{KL} \left( p_{g_\theta} \| p_{data} \right) - \mathrm{JSD} \left( p_{g_\theta} \| p_{data} \right) \Big] \Big|_{\boldsymbol{\theta}=\boldsymbol{\theta}_0}, \quad (8)$$

which gives simplified explanations of the training dynamics and the missing mode issue only when the discriminator meets certain optimality criteria. Our generalized result enables understanding of broader situations. For instance, when the discriminator distribution $q_{\phi_0}(y|\boldsymbol{x})$ gives uniform guesses, or when $p_{g_\theta} = p_{data}$ that is indistinguishable by the discriminator, the gradients of the KL and JSD terms in Eq.(6) cancel out, which stops the generator learning.

**InfoGAN**    Chen et al. (2016) developed InfoGAN which additionally recovers (part of) the latent code $\boldsymbol{z}$ given sample $\boldsymbol{x}$. This can straightforwardly be formulated in our framework by introducing an extra conditional $q_\eta(\boldsymbol{z}|\boldsymbol{x}, y)$ parameterized by $\boldsymbol{\eta}$. As discussed above, GANs assume a degenerated code space for real examples, thus $q_\eta(\boldsymbol{z}|\boldsymbol{x}, y = 1)$ is fixed without free parameters to learn, and $\boldsymbol{\eta}$ is only associated to $y = 0$. The InfoGAN is then recovered by combining $q_\eta(\boldsymbol{z}|\boldsymbol{x}, y)$ with $q_\phi(y|\boldsymbol{x})$ in Eq.(3) to perform full reconstruction of both $\boldsymbol{z}$ and $y$:

$$\begin{aligned}
\max_{\boldsymbol{\phi}} \mathcal{L}_\phi &= \mathbb{E}_{p_\theta(\boldsymbol{x}|y)p(y)} \left[ \log q_\eta(\boldsymbol{z}|\boldsymbol{x}, y) q_\phi(y|\boldsymbol{x}) \right] \\
\max_{\boldsymbol{\theta},\boldsymbol{\eta}} \mathcal{L}_{\theta,\eta} &= \mathbb{E}_{p_\theta(\boldsymbol{x}|y)p(y)} \left[ \log q_\eta(\boldsymbol{z}|\boldsymbol{x}, y) q_\phi^r(y|\boldsymbol{x}) \right].
\end{aligned} \quad (9)$$

Again, note that $\boldsymbol{z}$ is encapsulated in the implicit distribution $p_\theta(\boldsymbol{x}|y)$. The model is expressed as the schematic graphical model in Figure 1(d). Let $q^r(\boldsymbol{x}|\boldsymbol{z}, y) \propto q_{\eta_0}(\boldsymbol{z}|\boldsymbol{x}, y) q_{\phi_0}^r(y|\boldsymbol{x}) p_{\theta_0}(\boldsymbol{x})$ be the augmented "posterior", the result in the form of Lemma.1 still holds by adding $\boldsymbol{z}$-related conditionals:

$$\begin{aligned}
& \nabla_\theta \Big[ - \mathbb{E}_{p_\theta(\boldsymbol{x}|y)p(y)} \big[ \log q_{\eta_0}(\boldsymbol{z}|\boldsymbol{x}, y) q_{\phi_0}^r(y|\boldsymbol{x}) \big] \Big] \Big|_{\boldsymbol{\theta}=\boldsymbol{\theta}_0} = \\
& \nabla_\theta \Big[ \mathbb{E}_{p(y)} \big[ \mathrm{KL} \left( p_\theta(\boldsymbol{x}|y) \big\| q^r(\boldsymbol{x}|\boldsymbol{z}, y) \right) \big] - \mathrm{JSD} \left( p_\theta(\boldsymbol{x}|y = 0) \big\| p_\theta(\boldsymbol{x}|y = 1) \right) \Big] \Big|_{\boldsymbol{\theta}=\boldsymbol{\theta}_0},
\end{aligned} \quad (10)$$

The new formulation is also generally applicable to other GAN-related variants, such as Adversarial Autoencoder (Makhzani et al., 2015), Predictability Minimization (Schmidhuber, 1992), and cycleGAN (Zhu et al., 2017). In the supplements we provide interpretations of the above models.

## 3.3    VARIATIONAL AUTOENCODERS (VAEs)

We next explore the second family of deep generative modeling. The resemblance of GAN generator learning to variational inference (Lemma.1) suggests strong relations between VAEs (Kingma & Welling, 2013) and GANs. We build correspondence between them, and show that VAEs involve minimizing a KLD in an opposite direction, with a degenerated adversarial discriminator.

The conventional definition of VAEs is written as:

$$\max_{\boldsymbol{\theta},\boldsymbol{\eta}} \mathcal{L}_{\theta,\eta}^{\mathrm{vae}} = \mathbb{E}_{p_{data}(\boldsymbol{x})} \Big[ \mathbb{E}_{\tilde{q}_\eta(\boldsymbol{z}|\boldsymbol{x})} \left[ \log \tilde{p}_\theta(\boldsymbol{x}|\boldsymbol{z}) \right] - \mathrm{KL}(\tilde{q}_\eta(\boldsymbol{z}|\boldsymbol{x}) \| \tilde{p}(\boldsymbol{z})) \Big], \quad (11)$$

where $\tilde{p}_\theta(\boldsymbol{x}|\boldsymbol{z})$ is the generator, $\tilde{q}_\eta(\boldsymbol{z}|\boldsymbol{x})$ the inference model, and $\tilde{p}(\boldsymbol{z})$ the prior. The parameters to learn are intentionally denoted with the notations of corresponding modules in GANs. VAEs appear to differ from GANs greatly as they use only real examples and lack adversarial mechanism.

To connect to GANs, we assume a perfect discriminator $q_*(y|\boldsymbol{x})$ which always predicts $y = 1$ with probability 1 given real examples, and $y = 0$ given generated samples. Again, for notational simplicity, let $q_*^r(y|\boldsymbol{x}) = q_*(1 - y|\boldsymbol{x})$ be the reversed distribution.

**Lemma 2.** *Let $p_\theta(\boldsymbol{z}, y|\boldsymbol{x}) \propto p_\theta(\boldsymbol{x}|\boldsymbol{z}, y)p(\boldsymbol{z}|y)p(y)$. The VAE objective $\mathcal{L}_{\theta,\eta}^{vae}$ in Eq.(11) is equivalent to (omitting the constant scale factor 2):*

$$\begin{aligned}
\mathcal{L}_{\theta,\eta}^{vae} &= \mathbb{E}_{p_{\theta_0}(\boldsymbol{x})} \Big[ \mathbb{E}_{q_\eta(\boldsymbol{z}|\boldsymbol{x}, y) q_*^r(y|\boldsymbol{x})} \left[ \log p_\theta(\boldsymbol{x}|\boldsymbol{z}, y) \right] - KL \left( q_\eta(\boldsymbol{z}|\boldsymbol{x}, y) q_*^r(y|\boldsymbol{x}) \big\| p(\boldsymbol{z}|y)p(y) \right) \Big] \\
&= \mathbb{E}_{p_{\theta_0}(\boldsymbol{x})} \Big[ - KL \left( q_\eta(\boldsymbol{z}|\boldsymbol{x}, y) q_*^r(y|\boldsymbol{x}) \big\| p_\theta(\boldsymbol{z}, y|\boldsymbol{x}) \right) \Big].
\end{aligned} \quad (12)$$

Here most of the components have exact correspondences (and the same definitions) in GANs and InfoGAN (see Table 1), except that the generation distribution $p_\theta(\boldsymbol{x}|\boldsymbol{z}, y)$ differs slightly from its

| Components | ADA | GANs / InfoGAN | VAEs |
|---|---|---|---|
| $\boldsymbol{x}$ | features | data/generations | data/generations |
| $y$ | domain indicator | real/fake indicator | real/fake indicator (degenerated) |
| $\boldsymbol{z}$ | data examples | code vector | code vector |
| $p_\theta(\boldsymbol{x}|y)$ | feature distr. | [I] generator, Eq.4 | [G] $p_\theta(\boldsymbol{x}|\boldsymbol{z}, y)$, generator, Eq.13 |
| $q_\phi(y|\boldsymbol{x})$ | discriminator | [G] discriminator | [I] $q_*(y|\boldsymbol{x})$, discriminator (degenerated) |
| $q_\eta(\boldsymbol{z}|\boldsymbol{x}, y)$ | — | [G] infer net (InfoGAN) | [I] infer net |
| KLD to min | same as GANs | $KL\left(p_\theta(\boldsymbol{x}|y)\|q^r(\boldsymbol{x}|y)\right)$ | $KL\left(q_\eta(\boldsymbol{z}|\boldsymbol{x}, y)q_*^r(y|\boldsymbol{x})\|p_\theta(\boldsymbol{z}, y|\boldsymbol{x})\right)$ |

Table 1: Correspondence between different approaches in the proposed formulation. The label "[G]" in bold indicates the respective component is involved in the generative process within our interpretation, while "[I]" indicates inference process. This is also expressed in the schematic graphical models in Figure 1.

counterpart $p_\theta(\boldsymbol{x}|y)$ in Eq.(4) to additionally account for the uncertainty of generating $\boldsymbol{x}$ given $\boldsymbol{z}$:

$$p_\theta(\boldsymbol{x}|\boldsymbol{z}, y) = \begin{cases} \tilde{p}_\theta(\boldsymbol{x}|\boldsymbol{z}) & y = 0 \\ p_{data}(\boldsymbol{x}) & y = 1. \end{cases} \tag{13}$$

We provide the proof of Lemma 2 in the supplementary materials. Figure 1(e) shows the schematic graphical model of the new interpretation of VAEs, where the only difference from InfoGAN (Figure 1(d)) is swapping the solid-line arrows (generative process) and dashed-line arrows (inference). As in GANs and InfoGAN, for the real example domain with $y = 1$, both $q_\eta(\boldsymbol{z}|\boldsymbol{x}, y = 1)$ and $p_\theta(\boldsymbol{x}|\boldsymbol{z}, y = 1)$ are constant distributions. Since given a fake sample $\boldsymbol{x}$ from $p_{\theta_0}(\boldsymbol{x})$, the *reversed* perfect discriminator $q_*^r(y|\boldsymbol{x})$ always predicts $y = 1$ with probability 1, the loss on fake samples is therefore degenerated to a constant, which blocks out fake samples from contributing to learning.

### 3.4 CONNECTING GANS AND VAES

Table 1 summarizes the correspondence between the approaches. Lemma.1 and Lemma.2 have revealed that both GANs and VAEs involve minimizing a KLD of respective inference and posterior distributions. In particular, GANs involve minimizing the $KL\left(p_\theta(\boldsymbol{x}|y)\|q^r(\boldsymbol{x}|y)\right)$ while VAEs the $KL\left(q_\eta(\boldsymbol{z}|\boldsymbol{x}, y)q_*^r(y|\boldsymbol{x})\|p_\theta(\boldsymbol{z}, y|\boldsymbol{x})\right)$. This exposes several new connections between the two model classes, each of which in turn leads to a set of existing research, or can inspire new research directions:

1) As discussed in Lemma.1, GANs now also relate to the variational inference algorithm as with VAEs, revealing a unified statistical view of the two classes. Moreover, the new perspective naturally enables many of the extensions of VAEs and vanilla variational inference algorithm to be transferred to GANs. We show an example in the next section.

2) The generator parameters $\boldsymbol{\theta}$ are placed in the opposite directions in the two KLDs. The asymmetry of KLD leads to distinct model behaviors. For instance, as discussed in Lemma.1, GANs are able to generate sharp images but tend to collapse to one or few modes of the data (i.e., mode missing). In contrast, the KLD of VAEs tends to drive generator to cover all modes of the data distribution but also small-density regions (i.e., mode covering), which usually results in blurred, implausible samples. This naturally inspires combination of the two KLD objectives to remedy the asymmetry. Previous works have explored such combinations, though motivated in different perspectives (Larsen et al., 2015; Che et al., 2017a; Pu et al., 2017). We discuss more details in the supplements.

3) VAEs within our formulation also include an adversarial mechanism as in GANs. The discriminator is perfect and degenerated, disabling generated samples to help with learning. This inspires activating the adversary to allow learning from samples. We present a simple possible way in the next section.

4) GANs and VAEs have inverted latent-visible treatments of $(\boldsymbol{z}, y)$ and $\boldsymbol{x}$, since we interpret sample generation in GANs as posterior inference. Such inverted treatments strongly relates to the symmetry of the sleep and wake phases in the wake-sleep algorithm, as presented shortly. In sec.6, we provide a more general discussion on a symmetric view of generation and inference.

### 3.5 CONNECTING TO WAKE SLEEP ALGORITHM (WS)

Wake-sleep algorithm (Hinton et al., 1995) was proposed for learning deep generative models such as Helmholtz machines (Dayan et al., 1995). WS consists of wake phase and sleep phase, which

optimize the generative model and inference model, respectively. We follow the above notations, and introduce new notations $\boldsymbol{h}$ to denote general latent variables and $\boldsymbol{\lambda}$ to denote general parameters. The wake sleep algorithm is thus written as:

$$
\begin{aligned}
\text{Wake}: \quad & \max_{\boldsymbol{\theta}} \mathbb{E}_{q_\lambda(\boldsymbol{h}|\boldsymbol{x})p_{data}(\boldsymbol{x})} \left[ \log p_\theta(\boldsymbol{x}|\boldsymbol{h}) \right] \\
\text{Sleep}: \quad & \max_{\boldsymbol{\lambda}} \mathbb{E}_{p_\theta(\boldsymbol{x}|\boldsymbol{h})p(\boldsymbol{h})} \left[ \log q_\lambda(\boldsymbol{h}|\boldsymbol{x}) \right].
\end{aligned}
\tag{14}
$$

Briefly, the wake phase updates the generator parameters $\boldsymbol{\theta}$ by fitting $p_\theta(\boldsymbol{x}|\boldsymbol{h})$ to the real data and hidden code inferred by the inference model $q_\lambda(\boldsymbol{h}|\boldsymbol{x})$. On the other hand, the sleep phase updates the parameters $\boldsymbol{\lambda}$ based on the generated samples from the generator.

The relations between WS and VAEs are clear in previous discussions (Bornschein & Bengio, 2014; Kingma & Welling, 2013). Indeed, WS was originally proposed to minimize the variational lower bound as in VAEs (Eq.11) with the sleep phase approximation (Hinton et al., 1995). Alternatively, VAEs can be seen as extending the wake phase. Specifically, if we let $\boldsymbol{h}$ be $\boldsymbol{z}$ and $\boldsymbol{\lambda}$ be $\boldsymbol{\eta}$, the wake phase objective recovers VAEs (Eq.11) in terms of generator optimization (i.e., optimizing $\boldsymbol{\theta}$). Therefore, we can see VAEs as generalizing the wake phase by also optimizing the inference model $q_\eta$, with additional prior regularization on code $\boldsymbol{z}$.

On the other hand, GANs closely resemble the sleep phase. To make this clearer, let $\boldsymbol{h}$ be $y$ and $\boldsymbol{\lambda}$ be $\boldsymbol{\phi}$. This results in a sleep phase objective identical to that of optimizing the discriminator $q_\phi$ in Eq.(3), which is to reconstruct $y$ given sample $\boldsymbol{x}$. We thus can view GANs as generalizing the sleep phase by also optimizing the generative model $p_\theta$ to reconstruct reversed $y$. InfoGAN (Eq.9) further extends the correspondence to reconstruction of latents $\boldsymbol{z}$.

# 4 TRANSFERRING TECHNIQUES

The new interpretation not only reveals the connections underlying the broad set of existing approaches, but also facilitates to exchange ideas and transfer techniques across the two classes of algorithms. For instance, existing enhancements on VAEs can straightforwardly be applied to improve GANs, and vice versa. This section gives two examples. Here we only outline the main intuitions and resulting models, while providing the details in the supplement materials.

## 4.1 IMPORTANCE WEIGHTED GANS (IWGAN)

Burda et al. (2015) proposed importance weighted autoencoder (IWAE) that maximizes a tighter lower bound on the marginal likelihood. Within our framework it is straightforward to develop importance weighted GANs by *copying* the derivations of IWAE side by side, with little adaptations. Specifically, the variational inference interpretation in Lemma.1 suggests GANs can be viewed as maximizing a lower bound of the marginal likelihood on $y$ (putting aside the negative JSD term):

$$
\log q(y) = \log \int p_\theta(\boldsymbol{x}|y) \frac{q_{\phi_0}^r(y|\boldsymbol{x})p_{\theta_0}(\boldsymbol{x})}{p_\theta(\boldsymbol{x}|y)} d\boldsymbol{x} \geq -\text{KL}(p_\theta(\boldsymbol{x}|y)\|q^r(\boldsymbol{x}|y)) + const.
\tag{15}
$$

Following (Burda et al., 2015), we can derive a tighter lower bound through a $k$-sample importance weighting estimate of the marginal likelihood. With necessary approximations for tractability, optimizing the tighter lower bound results in the following update rule for the generator learning:

$$
\nabla_\theta \mathcal{L}_k(y) = \mathbb{E}_{\boldsymbol{z}_1,\dots,\boldsymbol{z}_k \sim p(\boldsymbol{z}|y)} \left[ \sum\nolimits_{i=1}^{k} \widetilde{w_i} \nabla_\theta \log q_{\phi_0}^r(y|\boldsymbol{x}(\boldsymbol{z}_i, \boldsymbol{\theta})) \right].
\tag{16}
$$

As in GANs, only $y = 0$ (i.e., generated samples) is effective for learning parameters $\boldsymbol{\theta}$. Compared to the vanilla GAN update (Eq.(6)), the only difference here is the additional importance weight $\widetilde{w_i}$ which is the normalization of $w_i = \frac{q_{\phi_0}^r(y|\boldsymbol{x}_i)}{q_{\phi_0}(y|\boldsymbol{x}_i)}$ over $k$ samples. Intuitively, the algorithm assigns higher weights to samples that are more realistic and fool the discriminator better, which is consistent to IWAE that emphasizes more on code states providing better reconstructions. Hjelm et al. (2017); Che et al. (2017b) developed a similar sample weighting scheme for generator training, while their generator of discrete data depends on explicit conditional likelihood. In practice, the $k$ samples correspond to sample minibatch in standard GAN update. Thus the only computational cost added by the importance weighting method is by evaluating the weight for each sample, and is negligible. The discriminator is trained in the same way as in standard GANs.

|  | GAN | IWGAN |
| --- | --- | --- |
| MNIST | 8.34±.03 | **8.45±.04** |
| SVHN | 5.18±.03 | **5.34±.03** |
| CIFAR10 | 7.86±.05 | **7.89± .04** |

|  | CGAN | IWCGAN |
| --- | --- | --- |
| MNIST | 0.985±.002 | **0.987±.002** |
| SVHN | 0.797±.005 | **0.798±.006** |

|  | SVAE | AASVAE |
| --- | --- | --- |
| 1% | 0.9412 | **0.9425** |
| 10% | 0.9768 | **0.9797** |

Table 2: **Left**: Inception scores of GANs and the importance weighted extension. **Middle**: Classification accuracy of the generations by conditional GANs and the IW extension. **Right**: Classification accuracy of semi-supervised VAEs and the AA extension on MNIST test set, with 1% and 10% real labeled training data.

| Train Data Size | VAE | AA-VAE | CVAE | AA-CVAE | SVAE | AA-SVAE |
| --- | --- | --- | --- | --- | --- | --- |
| 1% | -122.89 | **-122.15** | -125.44 | **-122.88** | -108.22 | **-107.61** |
| 10% | -104.49 | **-103.05** | -102.63 | **-101.63** | -99.44 | **-98.81** |
| 100% | -92.53 | **-92.42** | -93.16 | **-92.75** | — | — |

Table 3: Variational lower bounds on MNIST test set, trained on 1%, 10%, and 100% training data, respectively. In the semi-supervised VAE (SVAE) setting, remaining training data are used for unsupervised training.

## 4.2 ADVERSARY ACTIVATED VAEs (AAVAE)

By Lemma.2, VAEs include a degenerated discriminator which blocks out generated samples from contributing to model learning. We enable adaptive incorporation of fake samples by activating the adversarial mechanism. Specifically, we replace the perfect discriminator $q_*(y|\boldsymbol{x})$ in VAEs with a discriminator network $q_\phi(y|\boldsymbol{x})$ parameterized with $\phi$, resulting in an adapted objective of Eq.(12):

$$\max_{\boldsymbol{\theta},\boldsymbol{\eta}} \mathcal{L}_{\theta,\eta}^{\mathrm{aavae}} = \mathbb{E}_{p_{\theta_0}(\boldsymbol{x})} \left[ \mathbb{E}_{q_\eta(\boldsymbol{z}|\boldsymbol{x},y)q_\phi^r(y|\boldsymbol{x})} \left[ \log p_\theta(\boldsymbol{x}|\boldsymbol{z},y) \right] - \mathrm{KL}(q_\eta(\boldsymbol{z}|\boldsymbol{x},y)q_\phi^r(y|\boldsymbol{x}) \| p(\boldsymbol{z}|y)p(y)) \right]. \quad (17)$$

As detailed in the supplementary material, the discriminator is trained in the same way as in GANs.

The activated discriminator enables an effective data selection mechanism. First, AAVAE uses not only real examples, but also generated samples for training. Each sample is weighted by the inverted discriminator $q_\phi^r(y|\boldsymbol{x})$, so that only those samples that resemble real data and successfully fool the discriminator will be incorporated for training. This is consistent with the importance weighting strategy in IWGAN. Second, real examples are also weighted by $q_\phi^r(y|\boldsymbol{x})$. An example receiving large weight indicates it is easily recognized by the discriminator, which means the example is hard to be simulated from the generator. That is, AAVAE emphasizes more on harder examples.

## 5 EXPERIMENTS

We conduct preliminary experiments to demonstrate the generality and effectiveness of the importance weighting (IW) and adversarial activating (AA) techniques. In this paper we do not aim at achieving state-of-the-art performance, but leave it for future work. In particular, we show the IW and AA extensions improve the standard GANs and VAEs, as well as several of their variants, respectively. We present the results here, and provide details of experimental setups in the supplements.

### 5.1 IMPORTANCE WEIGHTED GANS

We extend both vanilla GANs and class-conditional GANs (CGAN) with the IW method. The base GAN model is implemented with the DCGAN architecture and hyperparameter setting (Radford et al., 2015). Hyperparameters are not tuned for the IW extensions. We use MNIST, SVHN, and CIFAR10 for evaluation. For vanilla GANs and its IW extension, we measure inception scores (Salimans et al., 2016) on the generated samples. For CGANs we evaluate the accuracy of conditional generation (Hu et al., 2017) with a pre-trained classifier. Please see the supplements for more details.

Table 2, left panel, shows the inception scores of GANs and IW-GAN, and the middle panel gives the classification accuracy of CGAN and and its IW extension. We report the averaged results $\pm$ one standard deviation over 5 runs. The IW strategy gives consistent improvements over the base models.

### 5.2 ADVERSARY ACTIVATED VAES

We apply the AA method on vanilla VAEs, class-conditional VAEs (CVAE), and semi-supervised VAEs (SVAE) (Kingma et al., 2014), respectively. We evaluate on the MNIST data. We measure the variational lower bound on the test set, with varying number of real training examples. For each batch of real examples, AA extended models generate equal number of fake samples for training.

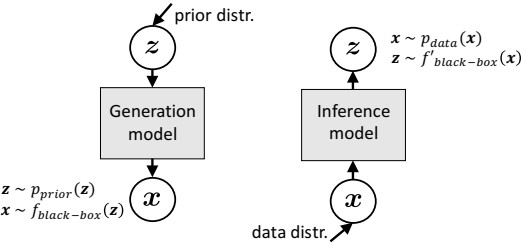

Figure 3: Symmetric view of generation and inference. There is little difference of the two processes in terms of formulation: with implicit distribution modeling, both processes only need to perform simulation through black-box neural transformations between the latent and visible spaces.

Table 3 shows the results of activating the adversarial mechanism in VAEs. Generally, larger improvement is obtained with smaller set of real training data. Table 2, right panel, shows the improved accuracy of AA-SVAE over the base semi-supervised VAE.

## 6 DISCUSSIONS: SYMMETRIC VIEW OF GENERATION AND INFERENCE

Our new interpretations of GANs and VAEs have revealed strong connections between them, and linked the emerging new approaches to the classic wake-sleep algorithm. The generality of the proposed formulation offers a unified statistical insight of the broad landscape of deep generative modeling, and encourages mutual exchange of techniques across research lines. One of the key ideas in our formulation is to interpret sample generation in GANs as performing posterior inference. This section provides a more general discussion of this point.

Traditional modeling approaches usually distinguish between latent and visible variables clearly and treat them in very different ways. One of the key thoughts in our formulation is that it is not necessary to make clear boundary between the two types of variables (and between generation and inference), but instead, treating them as a symmetric pair helps with modeling and understanding. For instance, we treat the generation space $x$ in GANs as latent, which immediately reveals the connection between GANs and adversarial domain adaptation, and provides a variational inference interpretation of the generation. A second example is the classic wake-sleep algorithm, where the wake phase reconstructs visibles conditioned on latents, while the sleep phase reconstructs latents conditioned on visibles (i.e., generated samples). Hence, visible and latent variables are treated in a completely symmetric manner.

- Empirical data distributions are usually implicit, i.e., easy to sample from but intractable for evaluating likelihood. In contrast, priors are usually defined as explicit distributions, amiable for likelihood evaluation.
- The complexity of the two distributions are different. Visible space is usually complex while latent space tends (or is designed) to be simpler.

However, the adversarial approach in GANs and other techniques such as density ratio estimation (Mohamed & Lakshminarayanan, 2016) and approximate Bayesian computation (Beaumont et al., 2002) have provided useful tools to bridge the gap in the first point. For instance, implicit generative models such as GANs require only simulation of the generative process without explicit likelihood evaluation, hence the prior distributions over latent variables are used in the same way as the empirical data distributions, namely, generating samples from the distributions. For explicit likelihood-based models, adversarial autoencoder (AAE) leverages the adversarial approach to allow implicit prior distributions over latent space. Besides, a few most recent work (Mescheder et al., 2017; Tran et al., 2017; Huszár, 2017; Rosca et al., 2017) extends VAEs by using implicit variational distributions as the inference model. Indeed, the reparameterization trick in VAEs already resembles construction of implicit variational distributions (as also seen in the derivations of IWGANs in Eq.37). In these algorithms, adversarial approach is used to replace intractable minimization of the KL divergence between implicit variational distributions and priors.

The second difference in terms of space complexity guides us to choose appropriate tools (e.g., adversarial approach v.s. reconstruction optimization, etc) to minimize the distance between distributions to learn and their targets. However, the tools chosen do not affect the underlying modeling mechanism.

For instance, VAEs and adversarial autoencoder both regularize the model by minimizing the distance between the variational posterior and certain prior, though VAEs choose KL divergence loss while AAE selects adversarial loss.

We can further extend the symmetric treatment of visible/latent $x/z$ pair to data/label $x/t$ pair, leading to a unified view of the generative and discriminative paradigms for unsupervised and semi-supervised learning. Specifically, conditional generative models create (data, label) pairs by generating data $x$ given label $t$. These pairs can be used for classifier training (Hu et al., 2017; Odena et al., 2017). In parallel, discriminative approaches such as knowledge distillation (Hinton et al., 2015; Hu et al., 2016) create (data, label) pairs by generating label $t$ conditioned on data $x$. With the symmetric view of $x$ and $t$ spaces, and neural network based black-box mappings across spaces, we can see the two approaches are essentially the same.

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

## A  ADVERSARIAL DOMAIN ADAPTATION (ADA)

ADA aims to transfer prediction knowledge learned from a source domain with labeled data to a target domain without labels, by learning domain-invariant features. Let $D_\phi(\boldsymbol{x}) = q_\phi(y|\boldsymbol{x})$ be the domain discriminator. The conventional formulation of ADA is as following:

$$
\begin{aligned}
&\max_\phi \mathcal{L}_\phi = \mathbb{E}_{\boldsymbol{x}=G_\theta(\boldsymbol{z}),\boldsymbol{z}\sim p(\boldsymbol{z}|y=1)} \left[\log D_\phi(\boldsymbol{x})\right] + \mathbb{E}_{\boldsymbol{x}=G_\theta(\boldsymbol{z}),\boldsymbol{z}\sim p(\boldsymbol{z}|y=0)} \left[\log(1 - D_\phi(\boldsymbol{x}))\right], \\
&\max_{\boldsymbol{\theta}} \mathcal{L}_\theta = \mathbb{E}_{\boldsymbol{x}=G_\theta(\boldsymbol{z}),\boldsymbol{z}\sim p(\boldsymbol{z}|y=1)} \left[\log(1 - D_\phi(\boldsymbol{x}))\right] + \mathbb{E}_{\boldsymbol{x}=G_\theta(\boldsymbol{z}),\boldsymbol{z}\sim p(\boldsymbol{z}|y=0)} \left[\log D_\phi(\boldsymbol{x})\right].
\end{aligned}
\tag{18}
$$

Further add the supervision objective of predicting label $t(\boldsymbol{z})$ of data $\boldsymbol{z}$ in the source domain, with a classifier $f_\omega(t|\boldsymbol{x})$ parameterized with $\boldsymbol{\pi}$:

$$
\max_{\boldsymbol{\omega},\boldsymbol{\theta}} \mathcal{L}_{\omega,\theta} = \mathbb{E}_{\boldsymbol{z}\sim p(\boldsymbol{z}|y=1)} \left[\log f_\omega(t(\boldsymbol{z})|G_\theta(\boldsymbol{z}))\right].
\tag{19}
$$

We then obtain the conventional formulation of adversarial domain adaptation used or similar in (Ganin et al., 2016; Purushotham et al., 2017).

## B  PROOF OF LEMMA 1

*Proof.*

$$
\begin{aligned}
&\mathbb{E}_{p_\theta(\boldsymbol{x}|y)p(y)} \left[\log q^r(y|\boldsymbol{x})\right] = \\
&- \mathbb{E}_{p(y)} \left[\mathrm{KL}\left(p_\theta(\boldsymbol{x}|y)\|q^r(\boldsymbol{x}|y)\right) - \mathrm{KL}(p_\theta(\boldsymbol{x}|y)\|p_{\theta_0}(\boldsymbol{x}))\right],
\end{aligned}
\tag{20}
$$

where

$$
\begin{aligned}
&\mathbb{E}_{p(y)} \left[\mathrm{KL}(p_\theta(\boldsymbol{x}|y)\|p_{\theta_0}(\boldsymbol{x}))\right] \\
&= p(y=0) \cdot \mathrm{KL}\left(p_\theta(\boldsymbol{x}|y=0)\|\frac{p_{\theta_0}(\boldsymbol{x}|y=0) + p_{\theta_0}(\boldsymbol{x}|y=1)}{2}\right) \\
&+ p(y=1) \cdot \mathrm{KL}\left(p_\theta(\boldsymbol{x}|y=1)\|\frac{p_{\theta_0}(\boldsymbol{x}|y=0) + p_{\theta_0}(\boldsymbol{x}|y=1)}{2}\right).
\end{aligned}
\tag{21}
$$

Note that $p_\theta(\boldsymbol{x}|y=0) = p_{g_\theta}(\boldsymbol{x})$, and $p_\theta(\boldsymbol{x}|y=1) = p_{data}(\boldsymbol{x})$. Let $p_{M_\theta} = \frac{p_{g_\theta}+p_{data}}{2}$. Eq.(21) can be simplified as:

$$
\mathbb{E}_{p(y)} \left[\mathrm{KL}(p_\theta(\boldsymbol{x}|y)\|p_{\theta_0}(\boldsymbol{x}))\right] = \frac{1}{2}\mathrm{KL}\left(p_{g_\theta}\|p_{M_{\theta_0}}\right) + \frac{1}{2}\mathrm{KL}\left(p_{data}\|p_{M_{\theta_0}}\right).
\tag{22}
$$

On the other hand,

$$
\begin{aligned}
\mathrm{JSD}(p_{g_\theta}\|p_{data}) &= \frac{1}{2}\mathbb{E}_{p_{g_\theta}} \left[\log \frac{p_{g_\theta}}{p_{M_\theta}}\right] + \frac{1}{2}\mathbb{E}_{p_{data}} \left[\log \frac{p_{data}}{p_{M_\theta}}\right] \\
&= \frac{1}{2}\mathbb{E}_{p_{g_\theta}} \left[\log \frac{p_{g_\theta}}{p_{M_{\theta_0}}}\right] + \frac{1}{2}\mathbb{E}_{p_{g_\theta}} \left[\log \frac{p_{M_{\theta_0}}}{p_{M_\theta}}\right] \\
&\quad + \frac{1}{2}\mathbb{E}_{p_{data}} \left[\log \frac{p_{data}}{p_{M_{\theta_0}}}\right] + \frac{1}{2}\mathbb{E}_{p_{data}} \left[\log \frac{p_{M_{\theta_0}}}{p_{M_\theta}}\right] \\
&= \frac{1}{2}\mathbb{E}_{p_{g_\theta}} \left[\log \frac{p_{g_\theta}}{p_{M_{\theta_0}}}\right] + \frac{1}{2}\mathbb{E}_{p_{data}} \left[\log \frac{p_{data}}{p_{M_{\theta_0}}}\right] + \mathbb{E}_{p_{M_\theta}} \left[\log \frac{p_{M_{\theta_0}}}{p_{M_\theta}}\right] \\
&= \frac{1}{2}\mathrm{KL}\left(p_{g_\theta}\|p_{M_{\theta_0}}\right) + \frac{1}{2}\mathrm{KL}\left(p_{data}\|p_{M_{\theta_0}}\right) - \mathrm{KL}\left(p_{M_\theta}\|p_{M_{\theta_0}}\right).
\end{aligned}
\tag{23}
$$

Note that

$$
\nabla_\theta \mathrm{KL}\left(p_{M_\theta}\|p_{M_{\theta_0}}\right) |_{\theta=\theta_0} = 0.
\tag{24}
$$

Taking derivatives of Eq.(22) w.r.t $\boldsymbol{\theta}$ at $\boldsymbol{\theta}_0$ we get

$$
\begin{aligned}
&\nabla_\theta \mathbb{E}_{p(y)} \left[\mathrm{KL}(p_\theta(\boldsymbol{x}|y)\|p_{\theta_0}(\boldsymbol{x}))\right] |_{\theta=\theta_0} \\
&= \nabla_\theta \left(\frac{1}{2}\mathrm{KL}\left(p_{g_\theta}\|p_{M_{\theta_0}}\right) |_{\theta=\theta_0} + \frac{1}{2}\mathrm{KL}\left(p_{data}\|p_{M_{\theta_0}}\right)\right) |_{\theta=\theta_0} \\
&= \nabla_\theta \mathrm{JSD}(p_{g_\theta}\|p_{data}) |_{\theta=\theta_0}.
\end{aligned}
\tag{25}
$$

Taking derivatives of the both sides of Eq.(20) at w.r.t $\boldsymbol{\theta}$ at $\boldsymbol{\theta}_0$ and plugging the last equation of Eq.(25), we obtain the desired results.  $\square$

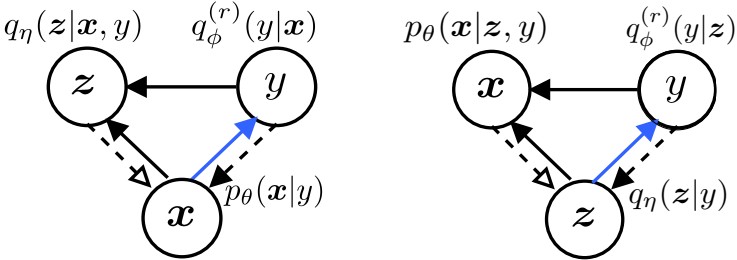

Figure 4: **Left:** Graphical model of InfoGAN. **Right:** Graphical model of Adversarial Autoencoder (AAE), which is obtained by swapping data $\boldsymbol{x}$ and code $\boldsymbol{z}$ in InfoGAN.

## C   PROOF OF JSD UPPER BOUND IN LEMMA 1

We show that, in Lemma.1 (Eq.6), the JSD term is upper bounded by the KL term, i.e.,

$$\text{JSD}(p_\theta(\boldsymbol{x}|y=0)\|p_\theta(\boldsymbol{x}|y=1)) \leq \mathbb{E}_{p(y)}\left[\text{KL}(p_\theta(\boldsymbol{x}|y)\|q^r(\boldsymbol{x}|y))\right]. \tag{26}$$

*Proof.* From Eq.(20), we have

$$\mathbb{E}_{p(y)}\left[\text{KL}(p_\theta(\boldsymbol{x}|y)\|p_{\theta_0}(\boldsymbol{x}))\right] \leq \mathbb{E}_{p(y)}\left[\text{KL}\left(p_\theta(\boldsymbol{x}|y)\|q^r(\boldsymbol{x}|y)\right)\right]. \tag{27}$$

From Eq.(22) and Eq.(23), we have

$$\text{JSD}(p_\theta(\boldsymbol{x}|y=0)\|p_\theta(\boldsymbol{x}|y=1)) \leq \mathbb{E}_{p(y)}\left[\text{KL}(p_\theta(\boldsymbol{x}|y)\|p_{\theta_0}(\boldsymbol{x}))\right]. \tag{28}$$

Eq.(27) and Eq.(28) lead to Eq.(26). $\qquad\qquad\qquad\qquad\qquad\qquad\qquad\qquad\qquad\square$

## D   SCHEMATIC GRAPHICAL MODELS AND AAE/PM/CYCLEGAN

**Adversarial Autoencoder (AAE)** (Makhzani et al., 2015) can be obtained by swapping code variable $\boldsymbol{z}$ and data variable $\boldsymbol{x}$ of InfoGAN in the graphical model, as shown in Figure 4. To see this, we directly write down the objectives represented by the graphical model in the right panel, and show they are precisely the original AAE objectives proposed in (Makhzani et al., 2015). We present detailed derivations, which also serve as an example for how one can translate a graphical model representation to the mathematical formulations. Readers can do similarly on the schematic graphical models of GANs, InfoGANs, VAEs, and many other relevant variants and write down the respective objectives conveniently.

We stick to the notational convention in the paper that parameter $\boldsymbol{\theta}$ is associated with the distribution over $\boldsymbol{x}$, parameter $\boldsymbol{\eta}$ with the distribution over $\boldsymbol{z}$, and parameter $\phi$ with the distribution over $y$. Besides, we use $p$ to denote the distributions over $\boldsymbol{x}$, and $q$ the distributions over $\boldsymbol{z}$ and $y$.

From the graphical model, the inference process (dashed-line arrows) involves implicit distribution $q_\eta(\boldsymbol{z}|y)$ (where $\boldsymbol{x}$ is encapsulated). As in the formulations of GANs (Eq.4 in the paper) and VAEs (Eq.13 in the paper), $y=1$ indicates the real distribution we want to approximate and $y=0$ indicates the approximate distribution with parameters to learn. So we have

$$q_\eta(\boldsymbol{z}|y) = \begin{cases} q_\eta(\boldsymbol{z}|y=0) & y=0 \\ q(\boldsymbol{z}) & y=1, \end{cases} \tag{29}$$

where, as $\boldsymbol{z}$ is the hidden code, $q(\boldsymbol{z})$ is the prior distribution over $\boldsymbol{z}$[1], and the space of $\boldsymbol{x}$ is degenerated. Here $q_\eta(\boldsymbol{z}|y=0)$ is the implicit distribution such that

$$\boldsymbol{z} \sim q_\eta(\boldsymbol{z}|y=0) \quad\Longleftrightarrow\quad \boldsymbol{z} = E_\eta(\boldsymbol{x}), \ \boldsymbol{x} \sim p_{data}(\boldsymbol{x}), \tag{30}$$

---

[1]See section 6 of the paper for the detailed discussion on prior distributions of hidden variables and empirical distribution of visible variables

where $E_\eta(x)$ is a deterministic transformation parameterized with $\eta$ that maps data $x$ to code $z$. Note that as $x$ is a visible variable, the pre-fixed distribution of $x$ is the empirical data distribution.

On the other hand, the generative process (solid-line arrows) involves $p_\theta(x|z, y)q_\phi^{(r)}(y|z)$ (here $q^{(r)}$ means we will swap between $q^r$ and $q$). As the space of $x$ is degenerated given $y = 1$, thus $p_\theta(x|z, y)$ is fixed without parameters to learn, and $\theta$ is only associated to $y = 0$.

With the above components, we maximize the log likelihood of the generative distributions $\log p_\theta(x|z, y)q_\phi^{(r)}(y|z)$ conditioning on the variable $z$ inferred by $q_\eta(z|y)$. Adding the prior distributions, the objectives are then written as

$$
\begin{aligned}
\max_\phi \mathcal{L}_\phi &= \mathbb{E}_{q_\eta(z|y)p(y)} \left[ \log p_\theta(x|z, y)q_\phi(y|z) \right] \\
\max_{\theta, \eta} \mathcal{L}_{\theta, \eta} &= \mathbb{E}_{q_\eta(z|y)p(y)} \left[ \log p_\theta(x|z, y)q_\phi^r(y|z) \right].
\end{aligned}
\tag{31}
$$

Again, the only difference between the objectives of $\phi$ and $\{\theta, \eta\}$ is swapping between $q_\phi(y|z)$ and its reverse $q_\phi^r(y|z)$.

To make it clearer that Eq.(31) is indeed the original AAE proposed in (Makhzani et al., 2015), we transform $\mathcal{L}_\phi$ as

$$
\begin{aligned}
\max_\phi \mathcal{L}_\phi &= \mathbb{E}_{q_\eta(z|y)p(y)} \left[ \log q_\phi(y|z) \right] \\
&= \frac{1}{2}\mathbb{E}_{q_\eta(z|y=0)} \left[ \log q_\phi(y = 0|z) \right] + \frac{1}{2}\mathbb{E}_{q_\eta(z|y=1)} \left[ \log q_\phi(y = 1|z) \right] \\
&= \frac{1}{2}\mathbb{E}_{z=E_\eta(x), x \sim p_{data}(x)} \left[ \log q_\phi(y = 0|z) \right] + \frac{1}{2}\mathbb{E}_{z \sim q(z)} \left[ \log q_\phi(y = 1|z) \right].
\end{aligned}
\tag{32}
$$

That is, the discriminator with parameters $\phi$ is trained to maximize the accuracy of distinguishing the hidden code either sampled from the true prior $p(z)$ or inferred from observed data example $x$. The objective $\mathcal{L}_{\theta, \eta}$ optimizes $\theta$ and $\eta$ to minimize the reconstruction loss of observed data $x$ and at the same time to generate code $z$ that fools the discriminator. We thus get the conventional view of the AAE model.

**Predictability Minimization (PM)** (Schmidhuber, 1992) is the early form of adversarial approach which aims at learning code $z$ from data such that each unit of the code is hard to predict by the accompanying code predictor based on remaining code units. AAE closely resembles PM by seeing the discriminator as a special form of the code predictors.

**CycleGAN** (Zhu et al., 2017) is the model that learns to translate examples of one domain (e.g., images of horse) to another domain (e.g., images of zebra) and vice versa based on unpaired data. Let $x$ and $z$ be the variables of the two domains, then the objectives of AAE (Eq.31) is precisely the objectives that train the model to translate $x$ into $z$. The reversed translation is trained with the objectives of InfoGAN (Eq.9 in the paper), the symmetric counterpart of AAE.

## E    PROOF OF LEMME 2

*Proof.* For the reconstruction term:

$$
\begin{aligned}
&\mathbb{E}_{p_{\theta_0}(x)} \left[ \mathbb{E}_{q_\eta(z|x, y)q_*^r(y|x)} \left[ \log p_\theta(x|z, y) \right] \right] \\
&= \frac{1}{2}\mathbb{E}_{p_{\theta_0}(x|y=1)} \left[ \mathbb{E}_{q_\eta(z|x, y=0), y=0 \sim q_*^r(y|x)} \left[ \log p_\theta(x|z, y = 0) \right] \right] \\
&\quad + \frac{1}{2}\mathbb{E}_{p_{\theta_0}(x|y=0)} \left[ \mathbb{E}_{q_\eta(z|x, y=1), y=1 \sim q_*^r(y|x)} \left[ \log p_\theta(x|z, y = 1) \right] \right] \\
&= \frac{1}{2}\mathbb{E}_{p_{data}(x)} \left[ \mathbb{E}_{\tilde{q}_\eta(z|x)} \left[ \log \tilde{p}_\theta(x|z) \right] \right] + const,
\end{aligned}
\tag{33}
$$

where $y = 0 \sim q_*^r(y|x)$ means $q_*^r(y|x)$ predicts $y = 0$ with probability 1. Note that both $q_\eta(z|x, y = 1)$ and $p_\theta(x|z, y = 1)$ are constant distributions without free parameters to learn; $q_\eta(z|x, y = 0) = \tilde{q}_\eta(z|x)$, and $p_\theta(x|z, y = 0) = \tilde{p}_\theta(x|z)$.

For the KL prior regularization term:

$$\mathbb{E}_{p_{\theta_0}(\boldsymbol{x})}\left[\text{KL}(q_\eta(\boldsymbol{z}|\boldsymbol{x},y)q_*^r(y|\boldsymbol{x})\|p(\boldsymbol{z}|y)p(y))\right]$$

$$= \mathbb{E}_{p_{\theta_0}(\boldsymbol{x})}\left[\int q_*^r(y|\boldsymbol{x})\text{KL}\left(q_\eta(\boldsymbol{z}|\boldsymbol{x},y)\|p(\boldsymbol{z}|y)\right)dy + \text{KL}\left(q_*^r(y|\boldsymbol{x})\|p(y)\right)\right]$$

$$= \frac{1}{2}\mathbb{E}_{p_{\theta_0}(\boldsymbol{x}|y=1)}\left[\text{KL}\left(q_\eta(\boldsymbol{z}|\boldsymbol{x},y=0)\|p(\boldsymbol{z}|y=0)\right) + const\right] + \frac{1}{2}\mathbb{E}_{p_{\theta_0}(\boldsymbol{x}|y=1)}\left[const\right]$$

$$= \frac{1}{2}\mathbb{E}_{p_{data}(\boldsymbol{x})}\left[\text{KL}(\tilde{q}_\eta(\boldsymbol{z}|\boldsymbol{x})\|\tilde{p}(\boldsymbol{z}))\right].$$

(34)

Combining Eq.(33) and Eq.(34) we recover the conventional VAE objective in Eq.(7) in the paper. □

## F   VAE/GAN JOINT MODELS FOR MODE MISSING/COVERING

Previous works have explored combination of VAEs and GANs. This can be naturally motivated by the asymmetric behaviors of the KL divergences that the two algorithms aim to optimize respectively. Specifically, the VAE/GAN joint models (Larsen et al., 2015; Pu et al., 2017) that improve the sharpness of VAE generated images can be alternatively motivated by remedying the mode covering behavior of the KLD in VAEs. That is, the KLD tends to drive the generative model to cover all modes of the data distribution as well as regions with small values of $p_{data}$, resulting in blurred, implausible samples. Incorporation of GAN objectives alleviates the issue as the inverted KL enforces the generator to focus on meaningful data modes. From the other perspective, augmenting GANs with VAE objectives helps addressing the mode missing problem, which justifies the intuition of (Che et al., 2017a).

## G   IMPORTANCE WEIGHTED GANS (IWGAN)

From Eq.(6) in the paper, we can view GANs as maximizing a lower bound of the "marginal log-likelihood" on $y$:

$$\log q(y) = \log \int p_\theta(\boldsymbol{x}|y)\frac{q^r(y|\boldsymbol{x})p_{\theta_0}(\boldsymbol{x})}{p_\theta(\boldsymbol{x}|y)}d\boldsymbol{x}$$

$$\geq \int p_\theta(\boldsymbol{x}|y)\log\frac{q^r(y|\boldsymbol{x})p_{\theta_0}(\boldsymbol{x})}{p_\theta(\boldsymbol{x}|y)}d\boldsymbol{x}$$

$$= -\text{KL}(p_\theta(\boldsymbol{x}|y)\|q^r(\boldsymbol{x}|y)) + const.$$

(35)

We can apply the same importance weighting method as in IWAE (Burda et al., 2015) to derive a tighter bound.

$$\log q(y) = \log \mathbb{E}\left[\frac{1}{k}\sum_{i=1}^{k}\frac{q^r(y|\boldsymbol{x}_i)p_{\theta_0}(\boldsymbol{x}_i)}{p_\theta(\boldsymbol{x}_i|y)}\right]$$

$$\geq \mathbb{E}\left[\log\frac{1}{k}\sum_{i=1}^{k}\frac{q^r(y|\boldsymbol{x}_i)p_{\theta_0}(\boldsymbol{x}_i)}{p_\theta(\boldsymbol{x}_i|y)}\right]$$

$$= \mathbb{E}\left[\log\frac{1}{k}\sum_{i=1}^{k}w_i\right]$$

$$:= \mathcal{L}_k(y)$$

(36)

where we have denoted $w_i = \frac{q^r(y|\boldsymbol{x}_i)p_{\theta_0}(\boldsymbol{x}_i)}{p_\theta(\boldsymbol{x}_i|y)}$, which is the unnormalized importance weight. We recover the lower bound of Eq.(35) when setting $k = 1$.

To maximize the importance weighted lower bound $\mathcal{L}_k(y)$, we take the derivative w.r.t $\boldsymbol{\theta}$ and apply the reparameterization trick on samples $\boldsymbol{x}$:

$$\nabla_\theta\mathcal{L}_k(y) = \nabla_\theta\mathbb{E}_{\boldsymbol{x}_1,\dots,\boldsymbol{x}_k}\left[\log\frac{1}{k}\sum_{i=1}^{k}w_i\right] = \mathbb{E}_{\boldsymbol{z}_1,\dots,\boldsymbol{z}_k}\left[\nabla_\theta\log\frac{1}{k}\sum_{i=1}^{k}w(y,\boldsymbol{x}(\boldsymbol{z}_i,\boldsymbol{\theta}))\right]$$

$$= \mathbb{E}_{\boldsymbol{z}_1,\dots,\boldsymbol{z}_k}\left[\sum_{i=1}^{k}\widetilde{w_i}\nabla_\theta\log w(y,\boldsymbol{x}(\boldsymbol{z}_i,\boldsymbol{\theta}))\right],$$

(37)

where $\widetilde{w_i} = w_i / \sum_{i=1}^{k} w_i$ are the normalized importance weights. We expand the weight at $\boldsymbol{\theta} = \boldsymbol{\theta}_0$

$$w_i|_{\theta=\theta_0} = \frac{q^r(y|\boldsymbol{x}_i)p_{\theta_0}(\boldsymbol{x}_i)}{p_\theta(\boldsymbol{x}_i|y)} = q^r(y|\boldsymbol{x}_i)\frac{\frac{1}{2}p_{\theta_0}(\boldsymbol{x}_i|y=0) + \frac{1}{2}p_{\theta_0}(\boldsymbol{x}_i|y=1)}{p_{\theta_0}(\boldsymbol{x}_i|y)}|_{\theta=\theta_0}. \tag{38}$$

The ratio of $p_{\theta_0}(\boldsymbol{x}_i|y=0)$ and $p_{\theta_0}(\boldsymbol{x}_i|y=1)$ is intractable. Using the Bayes' rule and approximating with the discriminator distribution, we have

$$\frac{p(\boldsymbol{x}|y=0)}{p(\boldsymbol{x}|y=1)} = \frac{p(y=0|\boldsymbol{x})p(y=1)}{p(y=1|\boldsymbol{x})p(y=0)} \approx \frac{q(y=0|\boldsymbol{x})}{q(y=1|\boldsymbol{x})}. \tag{39}$$

Plug Eq.(39) into the above we have

$$w_i|_{\theta=\theta_0} \approx \frac{q^r(y|\boldsymbol{x}_i)}{q(y|\boldsymbol{x}_i)}. \tag{40}$$

In Eq.(37), the derivative $\nabla_\theta \log w_i$ is

$$\nabla_\theta \log w(y, \boldsymbol{x}(\boldsymbol{z}_i, \boldsymbol{\theta})) = \nabla_\theta \log q^r(y|\boldsymbol{x}(\boldsymbol{z}_i, \boldsymbol{\theta})) + \nabla_\theta \log \frac{p_{\theta_0}(\boldsymbol{x}_i)}{p_\theta(\boldsymbol{x}_i|y)}. \tag{41}$$

The second term in the RHS of the equation is intractable as it involves evaluating the likelihood of implicit distributions. However, if we take $k = 1$, it can be shown that

$$
\begin{aligned}
&- \mathbb{E}_{p(y)p(\boldsymbol{z}|y)} \left[ \nabla_\theta \log \frac{p_{\theta_0}(\boldsymbol{x}(\boldsymbol{z}, \boldsymbol{\theta}))}{p_\theta(\boldsymbol{x}(\boldsymbol{z}, \boldsymbol{\theta})|y)}|_{\theta=\theta_0} \right] \\
&= - \nabla_\theta \frac{1}{2} \mathbb{E}_{p_\theta(\boldsymbol{x}|y=0)} \left[ \frac{p_{\theta_0}(\boldsymbol{x})}{p_\theta(\boldsymbol{x}|y=0)} \right] + \frac{1}{2} \mathbb{E}_{p_\theta(\boldsymbol{x}|y=1)} \left[ \frac{p_{\theta_0}(\boldsymbol{x})}{p_\theta(\boldsymbol{x}|y=1)} \right] |_{\theta=\theta_0} \\
&= \nabla_\theta \mathrm{JSD}(p_{g_\theta}(\boldsymbol{x}) \| p_{data}(\boldsymbol{x}))|_{\theta=\theta_0},
\end{aligned}
\tag{42}
$$

where the last equation is based on Eq.(23). That is, the second term in the RHS of Eq.(41) is (when $k = 1$) indeed the gradient of the JSD, which is subtracted away in the standard GANs as shown in Eq.(6) in the paper. We thus follow the standard GANs and also remove the second term even when $k > 1$. Therefore, the resulting update rule for the generator parameter $\boldsymbol{\theta}$ is

$$\nabla_\theta \mathcal{L}_k(y) = \mathbb{E}_{\boldsymbol{z}_1,...,\boldsymbol{z}_k \sim p(\boldsymbol{z}|y)} \left[ \sum_{i=1}^{k} \widetilde{w_i} \nabla_\theta \log q_{\phi_0}^r(y|\boldsymbol{x}(\boldsymbol{z}_i, \boldsymbol{\theta})) \right]. \tag{43}$$

## H  ADVERSARY ACTIVATED VAES (AAVAE)

In our formulation, VAEs include a degenerated adversarial discriminator which blocks out generated samples from contributing to model learning. We enable adaptive incorporation of fake samples by activating the adversarial mechanism. Again, derivations are straightforward by making symbolic analog to GANs.

We replace the perfect discriminator $q_*(y|\boldsymbol{x})$ in vanilla VAEs with the discriminator network $q_\phi(y|\boldsymbol{x})$ parameterized with $\phi$ as in GANs, resulting in an adapted objective of Eq.(12) in the paper:

$$\max_{\boldsymbol{\theta},\boldsymbol{\eta}} \mathcal{L}_{\theta,\eta}^{\mathrm{aavae}} = \mathbb{E}_{p_{\theta_0}(\boldsymbol{x})} \left[ \mathbb{E}_{q_\eta(\boldsymbol{z}|\boldsymbol{x},y)q_\phi^r(y|\boldsymbol{x})} \left[ \log p_\theta(\boldsymbol{x}|\boldsymbol{z},y) \right] - \mathrm{KL}(q_\eta(\boldsymbol{z}|\boldsymbol{x},y)q_\phi^r(y|\boldsymbol{x}) \| p(\boldsymbol{z}|y)p(y)) \right]. \tag{44}$$

The form of Eq.(44) is precisely symmetric to the objective of InfoGAN in Eq.(9) with the additional KL prior regularization. Before analyzing the effect of adding the learnable discriminator, we first look at how the discriminator is learned. In analog to GANs in Eq.(3) and InfoGANs in Eq.(9), the objective of optimizing $\phi$ is obtained by simply replacing the inverted distribution $q_\phi^r(y|\boldsymbol{x})$ with $q_\phi(y|\boldsymbol{x})$:

$$\max_\phi \mathcal{L}_\phi^{\mathrm{aavae}} = \mathbb{E}_{p_{\theta_0}(\boldsymbol{x})} \left[ \mathbb{E}_{q_\eta(\boldsymbol{z}|\boldsymbol{x},y)q_\phi(y|\boldsymbol{x})} \left[ \log p_\theta(\boldsymbol{x}|\boldsymbol{z},y) \right] - \mathrm{KL}(q_\eta(\boldsymbol{z}|\boldsymbol{x},y)q_\phi(y|\boldsymbol{x}) \| p(\boldsymbol{z}|y)p(y)) \right]. \tag{45}$$

Intuitively, the discriminator is trained to distinguish between real and fake instances by predicting appropriate $y$ that selects the components of $q_\eta(\boldsymbol{z}|\boldsymbol{x},y)$ and $p_\theta(\boldsymbol{x}|\boldsymbol{z},y)$ to best reconstruct $\boldsymbol{x}$. The difficulty of Eq.(45) is that $p_\theta(\boldsymbol{x}|\boldsymbol{z},y=1) = p_{data}(\boldsymbol{x})$ is an implicit distribution which is intractable for likelihood evaluation. We thus use the alternative objective as in GANs to train a binary classifier:

$$\max_\phi \mathcal{L}_\phi^{\mathrm{aavae}} = \mathbb{E}_{p_\theta(\boldsymbol{x}|\boldsymbol{z},y)p(\boldsymbol{z}|y)p(y)} \left[ \log q_\phi(y|\boldsymbol{x}) \right]. \tag{46}$$

# I EXPERIMENTS

## I.1 IMPORTANCE WEIGHTED GANS

We extend both vanilla GANs and class-conditional GANs (CGAN) with the importance weighting method. The base GAN model is implemented with the DCGAN architecture and hyperparameter setting (Radford et al., 2015). We do not tune the hyperparameters for the importance weighted extensions. We use MNIST, SVHN, and CIFAR10 for evaluation. For vanilla GANs and its IW extension, we measure inception scores (Salimans et al., 2016) on the generated samples. We train deep residual networks provided in the tensorflow library as evaluation networks, which achieve inception scores of $9.09$, $6.55$, and $8.77$ on the test sets of MNIST, SVHN, and CIFAR10, respectively. For conditional GANs we evaluate the accuracy of conditional generation (Hu et al., 2017). That is, we generate samples given class labels, and then use the pre-trained classifier to predict class labels of the generated samples. The accuracy is calculated as the percentage of the predictions that match the conditional labels. The evaluation networks achieve accuracy of $0.990$ and $0.902$ on the test sets of MNIST and SVHN, respectively.

## I.2 ADVERSARY ACTIVATED VAES

We apply the adversary activating method on vanilla VAEs, class-conditional VAEs (CVAE), and semi-supervised VAEs (SVAE) (Kingma et al., 2014). We evaluate on the MNIST data. The generator networks have the same architecture as the generators in GANs in the above experiments, with sigmoid activation functions on the last layer to compute the means of Bernoulli distributions over pixels. The inference networks, discriminators, and the classifier in SVAE share the same architecture as the discriminators in the GAN experiments.

We evaluate the lower bound value on the test set, with varying number of real training examples. For each minibatch of real examples we generate equal number of fake samples for training. In the experiments we found it is generally helpful to smooth the discriminator distributions by setting the temperature of the output sigmoid function larger than 1. This basically encourages the use of fake data for learning. We select the best temperature from $\{1, 1.5, 3, 5\}$ through cross-validation. We do not tune other hyperparameters for the adversary activated extensions.

Table 4 reports the full results of SVAE and AA-SVAE, with the average classification accuracy and standard deviations over 5 runs.

|  | 1% | 10% |
|---|---|---|
| SVAE | $0.9412\pm.0039$ | $0.9768\pm.0009$ |
| AASVAE | **$0.9425\pm.0045$** | **$0.9797\pm.0010$** |

Table 4: Classification accuracy of semi-supervised VAEs and the adversary activated extension on the MNIST test set, with varying size of real labeled training examples.

