# OpenReview forum: "On Unifying Deep Generative Models"
_ICLR.cc/2018/Conference — Accept (Poster)_

### Official Review · AnonReviewer3 · 2017-11-27
**Overall good perspective on GANs that connect them to other variational methods**

**Rating:** 7
**Confidence:** 4

**Review:**

Update 1/11/18:

I'm happy with the comments from the authors. I think the explanation of non-saturating vs saturating objective is nice, and I've increased the score.

Note though: I absolutely expect a revision at camera-ready if the paper gets accepted (we did not get one).

Original review:
The paper is overall a good contribution. The motivation / insights are interesting, the theory is correct, and the experiments support their claims.

I’m not sure I agree that this is “unifying” GANs and VAEs, rather it places them within the same graphical model perspective. This is very interesting and a valuable way of looking at things, but I don’t see this as reshaping how we think of or use GANs. Maybe a little less hype, a little more connection to other perspectives would be best. In particular, I’d hope the authors would talk a little more about f-GAN, as the variational lower-bound shown in this work is definitely related, though this work uniquely connects the GAN lower bound with VAE by introducing the intractable “posterior”, q(x | y).

Detailed comments:
P1: I see f-GAN as helping link adversarial learning with traditional likelihood-based methods, notably as a dual-formulation of the same problem. It seems like there should be some mention of this.

P2:
what does this mean: “generated samples from the generative model are not leveraged for model learning”. The wording is maybe a little confusing.

P5:
So here I think the connection to f-GAN is even clearer, but it isn’t stated explicitly in the paper: the discriminator defines a lower-bound for a divergence (in this case, the JSD), so it’s natural that there is an alternate formulation in terms of the posterior (as it is called in this work). As f-GAN is fairly well-known, not making this connection here I think isolates this work in a critical way that makes it seem that similar observations haven’t been made.

P6:
"which blocks out fake samples from contributing to learning”: this is an interesting way of thinking about this. One potential issue with VAEs / other MLE-based methods (such as teacher-forcing) is that it requires the model to stay “close” to the real data, while GANs do not have such a restriction. Would you care to comment on this?

P8:
I think both the Hjelm (BGAN) and Che (MaliGAN) are using these weights to address credit assignment with discrete data, but BGAN doesn’t use a MLE generator, as is claimed in this work.

General experimental comments:
Generally it looks like IWGAN and AA-VAE do as is claimed: IWGANs have better mode coverage (higher inception scores), while AA-VAEs have better likelihoods given that we’re using the generated samples as well as real data. This last one is a nice result, as it’s a general issue with RNNs (teacher forcing) and is why we need things like scheduled sampling to train on the free-running phase. Do you have any comments on this?

It would have been nice to show that this works on harder datasets (CelebA, LSUN, ImageNet).

---

> ### Author Response · Authors · 2017-12-15
> **Response to AnonReviewer3**
>
> By “unifying” we meant this work proposes a unified statistical view of VAEs and GANs. We will revise the title to avoid confusion.
>
> P1 & P5:
> We have discussed f-GAN (Nowozin et al., 2016) in the related work section. f-GAN and most previous works that analyze GANs are based on the `saturated` objective of GANs, i.e., min_G log(1-D(G(z))). In particular, f-GAN and a few other works showed that with this objective, GANs involve *minimizing a variational lower bound* of some f-divergence (Nowozin et al., 2016) or mutual information between x and y (the real/fake indicator) (Huszar et al., 2016; Li et al., 2016).
>
> In contrast, our work is based on the `non-saturated` objective of the original GAN, i.e., max_G log D(G(z)). The two objectives have the same fixed point solution, but the `non-saturated` one avoids the vanishing gradient issue of the `saturated` one, and is more widely-used in practice. However, very few formal analysis (e.g., Arjovsky & Bottou, 2017) has been done on the `non-saturated` objective. Our results in Lemma.1 is a generalization of the previous theorem in (Arjovsky & Bottou, 2017) by allowing non-optimal discriminators in the analysis (please see the last paragraph of P5 for more details).
>
> We will make these clearer in the revised version.
>
> P2 & P6:
> The sentence “generated samples from … learning” in P2 meant the same as “blocks out fake samples from contributing to learning” in P6 as the reviewer noted. We will polish the statement. Thanks for pointing it out.
>
> By the analysis in the paper and common empirical observations, GANs (which involve min KL(Q||P) ) suffer from mode missing issue. That is, the learned generation distribution tends to concentrate to few large modes of the real data distribution. In contrast, VAEs and other MLE-based methods (which involve min KL(P||Q) ) suffer from the issue of covering all data modes as well as small-density regions in-between. In this sense, GANs in practice are more “restricted” to stay close to data modes, and generate samples that are generally less diverse and more plausible.
>
> P8:
> BGAN for discrete data rephrases generator G as conditional distribution g(x|z), and evaluates the explicit conditional likelihood g(x|z) for training. BGAN for continuous data does not have such parameterization. We will update the statements to fix the issue. Thanks for pointing this out.
>
> Experiments:
> It can be very interesting to apply the techniques in AA-VAE to augment the MLE training of RNNs, which has not been explored in this paper. A related line of research is adversarial training of RNNs which applies a discriminator on the RNN samples. To our knowledge, such approaches suffer from optimization difficulty due to, e.g., the discrete nature of samples (e.g., text samples). In contrast, AA-VAE avoids the issue as generated samples are used in the same way as real data examples by maximizing the “likelihood” of good samples selected by the discriminator. We are happy to explore more in this direction in the future.
>
> This paper focuses mainly on establishing formal connections between GANs, VAEs, and other deep generative models through new formulations of them. Technique transfer between research lines, e.g., IWGAN and AA-VAE, serves to showcase the benefit of the unified statistical view. We will validate the new techniques on harder datasets as suggested, and show the results soon.

---

### Official Review · AnonReviewer1 · 2017-11-27
**Review of On Unifying Deep Generative Models**

**Rating:** 6
**Confidence:** 4

**Review:**

The authors develops a framework interpreting GAN algorithms as performing a form of variational inference on a generative model reconstructing an indicator variable of whether a sample is from the true of generative data distributions. Starting from the ‘non-saturated’ GAN loss the key result (lemma 1) shows that GANs minimizes the KL divergence between the generator(inference) distribution and a posterior distribution implicitly defined by the discriminator. I found the paper IWGAN and especially the AAVAE experiments quite interesting.  However the paper is also very dense and quite hard to follow at times - In general I think the paper would benefit from moving some content (like the wake-sleep part of the paper) to the appendix and concentrating more on the key results and a few more experiments as detailed in the comments / questions below.

Q1) What would happen if the KL-divergence minimizing loss proposed by Huszar (see e.g http://www.inference.vc/an-alternative-update-rule-for-generative-adversarial-networks/) was used instead of the “non-saturated” GAN loss - would the residial JSD terms in Lemma 1 cancel out then?

Q2) In Lemma 1 the negative JSD term looks a bit nasty to me e.g. in addition to KL divergence the GAN loss also maximises the JSD between the data and generative distributions. This JSD term acts in a somewhat opposite direction of the KL-divergence that we are interested in minimizing. Can the authors provide some more detailed comments / analysis on these two somewhat opposed terms - I find this quite important to include given the opposed direction of the JSD versus the KL term and that the JSD is ignored in e.g. section 4.1? secondly did the authors do any experiments on the the relative sizes of these two terms? I imagine it would be possible to perform some low-dimensional toy experiments where both terms were tractable to compute numerically?

Q3) I think the paper could benefit from some intuition / discussion of the posterior term q^r(x|y) in lemma 1 composed on the prior p_theta0(x) and discriminator q^r(y|x). The terms drops out nicely in math however i had a bit of a hard time wrapping my head around what minimizing the KL-divergence between this term and the inference distribution p(xIy). I know this is a kind of open ended question but i think it would greatly aid the reader in understanding the paper if more ‘guidance’ is provided instead of just writing “..by definition this is the posterior.’

Q4) In a similar vein to the above. It would be nice with some more discussion / definitions of the terms in Lemma 2. e.g what does “Here most of the components have exact correspondences (and the same definitions) in GANs and InfoGAN (see Table 1)” mean?

Q5) The authors state that there is ‘strong connections’ between VAEs and GANs. I agree that both (after some assumptions) both minimize a KL-divergence (table 1) however to me it is not obvious how strong this relation is. Could the authors provide some discussion / thoughts on this topic?

Overall i like this work but also feel that some aspects could be improved: My main concern is that a lot of the analysis hinges on the JSD term being insignificant, but the authors to my knowledge does but provide any prof / indications that this is actually true. Secondly I think the paper would greatly benefit from concentration on fewer topics (e.g. maybe drop the RW topic as it feels a bit like an appendix) and instead provide a more throughout discussion of the theory (lemma 1 + lemma 2) as well as some more experiments wrt JSD term.

---

> ### Author Response · Authors · 2017-12-15
> **Response to AnonReviewer1**
>
> Q1) Huszar proposed to optimize a loss that combines the `saturated` and `non-saturated` losses. We have cited and briefly discussed this work (Sønderby et al., 2017) in the related work section. As with most previous studies, the analysis of the combined loss in the blog and (Sønderby et al., 2017) is based on the assumption that the discriminator is near optimal. With this assumption, Lemma.1 is simplified to Eq.(8), and the residual JSD term cancels out with the combined loss. However, as discussed in the paper, Lemma.1 in general case (Eq.6) does not rely on the optimality assumptions of the discriminator which are usually unwarranted in practice. Thus, Lemma.1 can be seen as a generalization of previous results (Sønderby et al., 2017; Arjovsky and Bottou, 2017) to account for broader situations. Also, the JSD term does not cancel out even with the combined loss.
>
> Q2) We will update the paper to add more analysis of the JSD term. In particular, from the derivation of Lemma.1 in section C of the supplements, we can show the relative sizes of the KL and JSD term follow: JSD <= KL. Specifically, if we denote the RHS of Eq.(20) as -E_p(y) [ KL - KL_1 ], then from Eq.(20) we have KL_1 <= KL. From Eqs.(22) and (23), we further have JSD <= KL_1. We therefore have JSD <= KL_1 <= KL. That is, the JSD is upper-bounded by the KL, and intuitively, if the KL is sufficiently minimized, the magnitude of JSD will also decrease.
>
> Note that we did not mean that the JSD term is negligible. Indeed, most conclusions in the paper have taken into account the JSD. For example, JSD is *symmetric* (rather than insignificant) and will not affect the mode missing behavior of GANs endowed by the asymmetry of the KL. We have also noticed in the paper that the gradients of the JSD and the KLD cancel out when discriminator gives random guesses (e.g., when p_g=p_data). In the derivations of IWGAN in sections 4.1 and G, inspired from the JSD term in Lemma.1, we also subtracted away the 2nd term of RHS of Eq.(38) which equals the JSD when k=1. The approximation is necessary for computational tractability.
>
> Q3) Figure.2 and the second point (“Training dynamics”) under Lemma.1 give an intuitive illustration of the posterior distribution q^r(x|y). Intuitively, the posterior distribution is a mixture of p_data(x) and p_{g_\theta0}(x) with the mixing weights induced from the discriminator distribution q^r(y|x). Figure.2 illustrates how minimizing the KL divergence between the inference distribution and the posterior can push p_g towards p_data, and how mode missing can happen.
>
> Q4) We will add more definitions and explanations of the terms in Lemma.2. The key terms are listed in Table.1 to allow side-by-side comparison with the corresponding terms in GANs and InfoGANs. For example, the distribution q_\eta(z|x,y) in Lemma.2 precisely corresponds to the distribution q_\eta(z|x,y) in InfoGAN (defined in the text of Eq.(9)). We will make these clearer in the revised version. Thanks for the suggestion.
>
> Q5) By “strong” we meant the connections reveal multiple new perspectives of GANs and VAEs as well as a broad class of their variants. Most of the discussions are presented in section 3.4. For example, the reformulation of GANs links the adversarial approach to the classic Bayesian variational inference (VI) algorithm, which further opens up the opportunities of transferring the large volume of extensions of VI to the adversarial approach for improvement (e.g., the proposed IWGAN in the paper). Section 3.4 provides four examples of such new perspectives inspired by the new formulations and connections, each of which in turn leads to either an existing research direction or new broad discussions on deep generative modeling (e.g., section A). We hope this work can inspire even more insights and discussions on, e.g., formal relations of adversarial approaches and Bayesian methods, etc.

---

### Official Review · AnonReviewer2 · 2017-11-27
**Good paper**

**Rating:** 7
**Confidence:** 3

**Review:**

The paper provides a symmetric modeling perspective ("generation" and "inference" are just different naming, the underlying techniques can be exchanged) to unify existing deep generative models, particularly VAEs and GANs. Someone had to formally do this, and the paper did a good job in describing the new view (by borrowing the notations from adversarial domain adaptation), and demonstrating its benefits (by exchanging the techniques in different research lines). The connection to weak-sleep algorithm is also interesting. Overall this is a good paper and I have little to add to it.

One of the major conclusions is GANs and VAEs minimize the KL Divergence in opposite directions, thus are exposed to different issues, overspreading or missing modes. This has been noted and alleviated in [1].

Is it possible to revise the title of the paper to specifically reflect the proposed idea? Other papers have attempted to unify GAN and VAE from different perspectives [1,2].

[1] Symmetric variational autoencoder and connections to adversarial learning. arXiv:1709.01846
[2] Adversarial variational Bayes: Unifying variational autoencoders and generative adversarial networks. arXiv:1701.04722, 2017.


Minor: In Fig. 1, consider to make “(d)” bold to be consistent with other terms.

---

> ### Author Response · Authors · 2017-12-15
> **Response to AnonReviewer2**
>
> Thanks for the valuable and encouraging comments.
>
> - Our work is indeed a couple of months earlier than [1], and is discussed in [1]. The work of [1] focuses on alleviating the asymmetry of the KL Divergence minimized by VAEs. It discusses the connection of the new symmetric VAE variant and GANs, but does not reveal that GANs involve minimizing a KL Divergence in an opposite direction, nor focus on the underlying connections between original VAEs and GANs as we do.
>
> In section 3.4 point 2) and section F of the supplements, we discussed some existing work on alleviating the mode overspreading issue of VAEs by augmenting original VAE objective with GANs related objectives. The work of [1] falls into this category (though in [1] the symmetric VAE is motivated purely from VAEs perspective). We will include the discussion of [1] in the revised version.
>
> - Our work aims at developing a unified statistical view of VAEs and GANs through new formulations of them. The unified view provides a tool to analyze existing deep generative model research, and naturally enables technique transfer between research lines. This is different from other work [1,2] which combines VAE and GAN objectives to form a new model/algorithm instance. We acknowledge that a clearer, specific title can alleviate confusions. Thanks for the suggestion.

---

### Decision · Program_Chairs · 2018-01-29
**ICLR 2018 Conference Acceptance Decision**

**Decision:**

Accept (Poster)

**Comment:**

This is a thought-provoking paper that places GANs and VAEs in a single framework and, motivated by this perspective, proposes several novel extensions to them. The reviewers made several good suggestions for improving the paper and the authors are expected to make the revisions they promised. The current title of the paper is too general and should be changed to something more directly descriptive of the contents.